

# Machine-Learning blends of geomorphic descriptors: value and limitations for flood hazard assessment across large floodplains

Andrea Magnini[1], Michele Lombardi[2], Simone Persiano[1], Antonio Tirri[3], Francesco Lo Conti[3], and Attilio Castellarin[1]

[1]Department of Civil, Chemical, Environmental and Materials Engineering (DICAM), University of Bologna, Bologna, Italy
[2]Department of Computer Science and Engineering (DISI), University of Bologna, Bologna, Italy
[3]Leithà, Unipol Group, Milan and Bologna, Italy

**Correspondence:** Andrea Magnini (andrea.magnini@studio.unibo.it)

**Abstract.** Recent literature shows several examples of simplified approaches that perform flood hazard (FH) assessment and mapping across large geographical areas on the basis of fast-computing geomorphic descriptors. These approaches may consider a single index (univariate) or use a set of indices simultaneously (multivariate). What is the potential and accuracy of multivariate approaches relative to univariate ones? Can we effectively use these methods for extrapolation purposes, i.e. FH

assessment outside the region used for setting up the model? Our study addresses these open problems by considering two separate issues: (1) mapping flood-prone areas, and (2) predicting the expected water depth for a given inundation scenario. We blend seven geomorphic descriptors through Decision Tree models trained on target FH maps, referring to a large study area ($\sim 10^5$ km$^2$). We discuss the potential of multivariate approaches relative to the performance of a selected univariate model and on the basis of multiple extrapolation experiments, where models are tested outside their training region. Our results show

that multivariate approaches may (a) significantly enhance flood-prone area delineation (overall accuracy: 93%) relative to univariate ones (overall accuracy: 84%), (b) provide accurate predictions of expected inundation depths (determination coefficient $\sim 0.7$), and (c) produce encouraging results in extrapolation.

## 1 Introduction

Every year flood events worldwide cause vast economic losses, as well as heavy social and environmental impacts, which have

been steadily increasing over the last five decades (Jongman et al., 2014; Guha-Sapir et al., 2016), mainly because of the complex interaction between the intensification of extreme hydrological events due to climate change (e.g., Brunetti et al., 2002; Uboldi and Lussana, 2018) and anthropogenic pressure (i.e., land-use and land-cover modifications, see Di Baldassarre et al., 2013; Domeneghetti et al., 2015; Requena et al., 2017). Thus, nowadays, successful flood hazard mapping for flood hazard management is a major task for the whole scientific community (Alfieri et al., 2014; Dottori et al., 2016). Traditional methods

to assess fluvial flood hazard rely on hydrological and hydraulic numerical models. Improvement of these tools allows to simulate any scenario for different geometrical or hydrological conditions, obtaining very accurate results (Horritt and Bates, 2002; Costabile et al., 2012; Bellos and Tsakiris, 2016). However, a high amount of hydrologic and hydraulic input information is required to adequately describe the geometry and hydraulic behaviour of the system, thus considerable effort and computation





capacity are needed. Consequently, numerical models are unsuitable for large-scale applications and in data-scarce regions.

To overcome this issue, other mapping techniques have been proposed that take advantage of the wealth of topographic information contained in digital elevation models (DEMs): flood-related geomorphic descriptors (or features, or indices) can be derived from DEMs and used to obtain a measure of flood hazard.

The first DEM-based approaches proposed in the literature (see e.g., Williams et al., 2000; Noman et al., 2001; Dodov and Foufoula-Georgiou , 2006; Nardi et al., 2006; Manfreda et al., 2011, 2014, 2015; Samela et al., 2017; De Risi et al., 2018)

consider a single geomorphic index (these approaches will be referred to as univariate hereafter), which is used as a binary classifier to distinguish between flood-prone and flood-free areas through the definition of a threshold value. The optimal threshold value is identified by means of an iterative calibration procedure which optimizes the agreement of the binary map with a reference pre-existing flood hazard map obtained, e.g., from hydrological-hydraulic numerical simulations. Several authors (see e.g., Manfreda et al., 2015; Samela et al., 2017) highlight that the performance of the considered geomorphic index can change

according to the geographical context of the application. In particular, the descriptor named Geomorphic Flood Index (GFI; Samela et al., 2017) has been shown to have good effectiveness in mountainous as well as in predominantly flat areas, and thus has been used extensively by many authors for developing web-services, platforms, and GIS tools for flood-hazard mapping applications (Samela et al., 2018; Tavares da Costa et al., 2019). A second class of DEM-based approaches to be investigated can be named as multivariate, as they rely on the combination of different geomorphic descriptors (GDs). The relation be-

tween the combination of GDs and flood hazard can be searched through numerous statistical methods. Commonly, Machine Learning (ML; Breiman, 1984) models are used, often ensembled with Multi-Criteria Decision-Making (MCDM) techniques (Triantaphyllou et al., 2000; Ho et al., 2010). Some authors (Degiorgis et al., 2012; Gnecco et al., 2017) have tested a blend of GDs, while some others mixed these indices with information on land use, soil geology and climate, and compared different combination strategies (e.g., Wang et al., 2015; Lee et al., 2017; Khosravi et al., 2018; Arabameri et al., 2019; Janizadeh et al.,

2019; Costache et al., 2020). These studies suggest that data-driven flood hazard mapping has a remarkable potential. However, in most of the studies, the reference flood hazard information used to set up the models consists of a dataset of isolated historical events observed in the study area (Lee et al., 2017; Khosravi et al., 2018; Janizadeh et al., 2019; Arabameri et al., 2019; Costache et al., 2020), leading to case-specific prediction skills.

Important advantages of DEM-based flood hazard mapping methods are their flexibility and, in principle, their general applica-

50 bility to any flood-prone area where a reliable DEM is available, as well as their low computational costs relative to numerical models. However, two main drawbacks must be highlighted: first, DEM-based methods do not consider the water dynamics, and second, they need a pre-existing reliable reference flood hazard map, which may or may not be available for the area of interest. Overall, DEM-based models are very useful as preliminary flood hazard mapping tools in data-scarce contexts and in application to large areas, but cannot yet effectively substitute the traditional models, especially when detailed results are

55 required. Nevertheless, if a strong and reliable relation to derive flood hazard from GDs is obtained, the model could be easily applied in extrapolation to any region where the same relation is supposed to be valid (Tavares da Costa et al., 2020).

In this study, multivariate DEM-based flood hazard mapping is investigated. We consider a large study area ($10^5 km^2$) in Northern Italy, which is characterized by markedly different morphological, hydrological and climatic conditions. We use the $\sim 90m$





resolution, hydrologically-corrected, MERIT DEM (Yamazaki et al., 2017) for deriving a set of GDs. We then use decision

trees, a common machine learning technique (Hastie et al., 2009), for assessing flood hazard associated with a given probability of occurrence (i.e., return period) in terms of (a) delineation of flood-prone and flood-free areas, and (b) prediction of expected inundation water depth (as a measure for flood intensity). The simultaneous combination of the five following meaningful elements makes our study different from all previous works in literature. First, only strictly easy-to-retrieve, DEM-based GDs are used to assess flood hazard, in contrast with several studies in which also other information is considered (e.g., soil geology,

permeability, rainfall data, as in Wang et al., 2015; Lee et al., 2017; Khosravi et al., 2018; Arabameri et al., 2019; Janizadeh et al., 2019; Costache et al., 2020). Second, both generation of binary flood hazard maps and prediction of expected maximum inundation water depth are analyzed, setting up parallel models (different from, e.g., Faridani, et al., 2020; Hosseiny et al., 2020). Third, decision trees are trained using pre-existing flood hazard maps as target information, in contrast with the discontinuous datasets of historical events mostly used to train machine learning models for flood hazard estimation (Lee et al., 2017; Khos-

ravi et al., 2018; Janizadeh et al., 2019; Arabameri et al., 2019; Costache et al., 2020). Fourth, univariate geomorphological approach for identification of flood-prone/flood-free areas (i.e., GFI) is compared with the proposed multivariate approach (i.e., the combination of the blend of DEM-based GDs by means of decision trees): this allows to analyse the actual enhancement resulting from the use of multiple GDs. Fifth, predictive skill of the multivariate DEM-based flood hazard approach is assessed in extrapolation by applying models trained on specific geographical areas to different regions with dissimilar morphological

and/or hydrological features. This last aspect is highly important for possible future applications to data-scarce environments in extrapolation mode.

By assuming the overmentioned characteristics, this study aims to advance previous knowledge on the potential of ML techniques for combining GDs to derive accurate flood hazard maps across large geographical regions. More precisely, we want to investigate three main research questions: (1) can we profit from a blend of various GDs for flood hazard assessment and

80 mapping relative to a univariate approach? (2) Can we use simple ML techniques for effectively blending multiple GDs? (3) Are these techniques capable of providing a reliable assessment of flood hazard over large geographical areas when used in geographical extrapolation? What are the desired characteristics of the training region/watershed to make the trained model as general as possible?

## 2 Methods

The methodologies adopted in the present study aim to define models that estimate flood hazard output variables (i.e., flood-susceptibility and maximum expected water depth) by combining several selected DEM-derived input features, based on the availability of target information (i.e., flood hazard reference maps). To this aim, two major steps are considered: (1) the DEM-based terrain analysis used for retrieving the selected input features (i.e., geomorphic descriptors, or GDs); (2) the definition of predictive models, which in our study is based on training and testing decision trees (DTs). Consistent with the aims of

our study, we set up two different types of Decision Trees (DTs): Classifier DTs to solve the classification problem relative to flood-extent delineation, and Regressor DTs to solve the regression problem of water depth estimation. Classifier and Regressor



DTs use the same input information (i.e., GDs), but require different target flood hazard maps. The software we use to train the DTs is Scikit-learn (Pedregosa et al., 2011), open source library for Python 3.6 or later (Van Rossum et al., 1995).

## 2.1 DEM-processing to obtain geomorphic descriptors

Topographical rasterized information contained in DEMs can be used to extract GDs adopting several algorithms available in the literature (e.g., Tarboton et al., 1991). These descriptors vary spatially, assuming different values for different pixels within the domain, while being constant in time. They can be divided into two broad categories: (1) single features, if they represent simple terrain characteristics, and (2) composite indices, if they are derived based on a combination of different other indices. As input variables for the above-mentioned models, in our study we use the ground elevation in m a.s.l. itself (as retrieved from 100 the DEM) together with six GDs, the first three of which are single indices, while the remaining are composite:

1. Local slope (sd8), estimated for each cell as the maximum slope among the eight possible flow directions and computed as the ratio between the vertical and the horizontal differences

2. Horizontal distance from the nearest stream (D), defined as the length of the path that hydrologically connects each cell to the nearest cell of the river network

3. Height above the nearest drainage (HAND), defined as the vertical difference between a given cell and the hydrologically nearest cell belonging to the river network (Rennò et al., 2008)

4. Modified topographic index ($TI_m$), derived from the modification proposed by Manfreda et al. (2008) to the index originally introduced by Kirkby (1975), and defined as follows:

$$TI_m = \ln\left(\frac{a_d^n}{\tan(\beta)}\right) \tag{1}$$

where $a_d$ (m) is the drained area per unit contour length, $\tan(\beta)$ is the local gradient, $n$ is an exponent <1

5. Geomorphic flood index (GFI), defined as the ratio between the term $h_r$ and HAND. The numerator represents the water depth, computed in the hydrologically nearest stream section with a hydraulic scale relation ($h_r \cong bA_r^n$, where $A_r$ is the contributing area in the considered stream section), where coefficient $b$ and exponent $n$ can be appropriately estimated with calibration or taken from the literature (Nardi et al., 2006)

$$GFI = \ln\left(\frac{h_r}{HAND}\right) \tag{2}$$

6. Alternative version of the GFI, hereinafter referred to as local geomorphic flood index (LGFI), defined as:

$$LGFI = \ln\left(\frac{h_l}{HAND}\right) \tag{3}$$

where the water depth $h_l$ is computed with reference to the contributing area of the considered pixel





The choice of the above mentioned GDs is due to different reasons. First, previous studies (e.g., Manfreda et al., 2015;
Samela et al., 2017) clearly showed that D and HAND are the most descriptive single-feature indices for flood hazard mapping,
sufficiently accurate in mountainous regions, but still inadequate over predominantly flat areas, whereas, among composite
feature indices, GFI and LGFI show good performance in both the geographical contexts. Also, in several studies (e.g., Wang
et al., 2015; Lee et al., 2017; Khosravi et al., 2018; Janizadeh et al., 2019; Costache et al., 2020), elevation retrieved from DEM
shows to have a strong influence on flood occurrence. Slope appears to be the most important index in Khosravi et al. (2018)
and Costache et al. (2020), and among the most influent ones in Arabameri et al. (2019).

While the use of elevation, sd8, D, HAND, GFI and LGFI is well-established in previous literature, the adoption of $TI_m$ is
based on Manfreda et al. (2008), who highligthed a strong correlation between the index and the occurrence of inunvdation
events.

Finally, we believe that the selected set of GDs provides DT models with a rather exhaustive description of the study area
morphology. In fact, slope and $TI_m$ may influence the infiltration time, and consequently the runoff; elevation is not only
strongly linked to the runoff, but also to climatic conditions; D and HAND consider the horizontal and vertical proximity to
the river network, and GFI and LGFI combine this information with an estimation of the water depth in the nearest stream.
Overall, for the aim of a multivariate analysis, this combination should enable one to consider two comprehensive pieces of
information by looking into the morphology (i.e. elevation, sd8, $TI_m$) and hydrology (i.e., by accounting for the river network;
i.e. D, HAND, GFI, LGFI) of the study region.

## 2.2 Decision trees

Supervised ML models can be thought of as complex, parameterized functions that are trained to accomplish a specific task. In
so-called supervised learning, training algorithms determine the structure and parameters of the model, by observing a series
of examples, i.e. input-output pairs. Decision Trees (DTs) are very popular supervised ML techniques (Breiman, 1984; Hastie
et al., 2009), as they are very effective in solving many kinds of problem based on an easily-interpretable logic.

DTs search for a relation between input and target output by means of a recursive splitting, which is done through a set of
nodes organized in a tree structure. Being the input of a DT a vector of values for a fixed set of "attributes" (or "features"),
each node corresponds to a test to be performed on a single attribute in the input vector. Depending on the outcome of the tests
on the nodes, the data is forwarded to one of a set of "child" nodes (see Figure 1). Leaves are the last nodes; they are labeled
with an output value, such as a class or a number, that represents the tree output for the given input vector.

Training a Decision Tree consists in determining its structure, the test on each node, and the labels on the leaves. Most training
algorithms operate by recursively splitting the training set, measuring the quality of each partition with object functions that
reflect the degree of uniformity of the output values (see Sect. 4.4). Repeatedly, tests leading to the best partition are chosen,
and child nodes are created accordingly. When some termination criterion is reached, e.g. a set in the partition is perfectly
uniform or a maximum depth has been reached, the last nodes become leaves and they are labeled either with the most frequent
class value (discrete case) or with the average of the output values (numeric case).





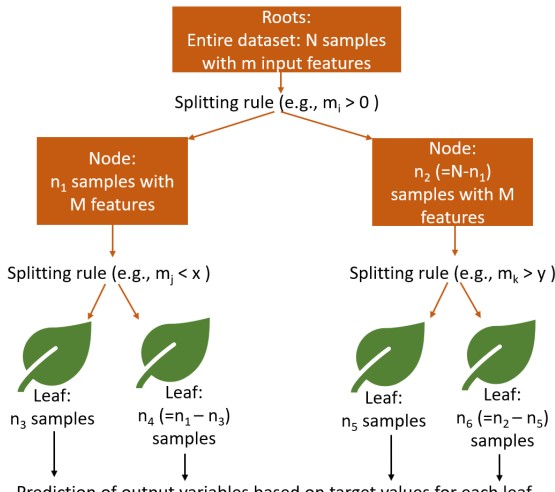

**Figure 1.** Exemplifying structure of a decision tree for a given dataset with N samples and M features

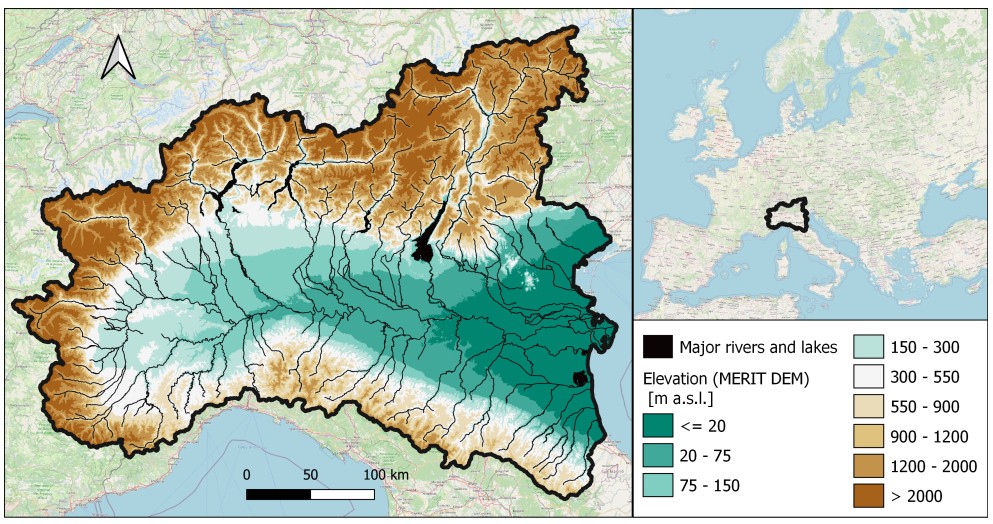

**Figure 2.** MERIT DEM for the study area, with major rivers and lakes marked in black (left); study area in the European context (right; map from ©OpenStreetMap, made available here under the Open Database License by OpenStreeMap Foundation)

## 3    Study area

The study area includes most of Northern Italy, and a little part of Switzerland, having a total extent of about $10^5 km^2$. Many different geographical subsystems can be found within this surface: the Alps, located in the North, lie in about $5 \cdot 10^4 km^2$, with average elevation of 2500 $ma.s.l.$ and a mainly rocky soil. This mountain range also hosts several big lakes, as Garda,





Maggiore and Iseo Lakes. The Apennines, in the southern portion, have lower altitudes than the Alps, and more permeable soils. The Po Valley, the largest floodplain in Italy, stretches from West to East, covering an area of about $4.6 \cdot 10^4 km^2$, going from the Alps and the Apennines to the Adriatic Sea (see Figure 2). The study area is mostly occupied by the river Po basin, that is the largest in Italy. Moreover, other important rivers are the Adige, Brenta, Reno and Bacchiglione.

Floods are a major problem in this region, both because of their high frequency over these large and predominantly flat areas and due to the presence of important industrial and agricultural assets and numerous cities (Persiano et al., 2020). The DEM used to represent the study area is the freely-available Multi-Error-Remover Improved-Terrain model (MERIT; see Yamazaki et al., 2017). This choice was made for two reasons. First, MERIT should be quite reliable for hydrological applications, as it is the product of several processing operations and corrections on previously available DEMs (i.e., NASA SRTM3 and

JAXA AW3D), some of which specifically addressing hydrological consistency (e.g. agreement between modelled and real stream-network). The second reason is that its resolution is 3 arcseconds, which corresponds to $\sim 90m$ at the equator. These characteristics enabled us to perform an accurate computation of geomorphic indices, while reducing the computational costs. Two different freely-available reference flood hazard maps have been used to train the ML models. The first, used for the classification problem (i.e., delineation of flood-extent), has been produced by the Italian Institute for Environmental Protection

and Research (ISPRA) to fulfill the Floods Directive of the European Parliament (2007/60/EC). This map (hereinafter referred to as PGRA P1) refers to a return period of about 500 years and comes from the merge of different hazard maps produced by local authorities, which explains its heterogeneity. Detailed flood hazard mapping characterizes some areas (e.g., see the northwestern portion of the study area in Figure 3), while lacking information affects other zones (e.g., see the northeastern portion of the study area in Figure 3). In the reminder of this study we term exhaustiveness the degree of detail by which

flood hazard is defined and captured for minor streams. The second map (see Figrue 4), used for the regression problem (i.e., estimation of water depth), was produced by the study from the Joint Research Centre (JRC) of the European Commission and refers to a return period of 100 years; it will be named JRC 100 in the remainder of the study. Differently from PGRA P1, JRC 100 provides information in terms of water depth and is uniform throughout the study area, yet evenly incomplete and less accurate for minor streams, as it comes from the merger of several numerical simulations, which considered only river

catchments with drainage area higher than $500km^2$ (see Dottori et al., 2016).

## 4   Framework of the analysis

This section provides an overview of the four macro-phases (see Figure 5) of our study: (1) DEM processing, for retrieving and computing the selected geomorphic descriptors; (2) configuration of the DTs, aimed at identifying a set structure for all subsequent analyses; (3) assessment of the DTs performances using training and test sets with the same statistical distribution;

(4) assessment of DTs predictive skill when used in geographical extrapolation. Results from phases 3 and 4 enable one to observe the model performances in multiple different applications, and to advance our knowledge on their value and limitations. In particular, phase (2) consists of four consecutive preliminary steps. The first step (a) is the definition of the conceptual scheme of the processing operations to set up the models, which is based on the following four points: (i) discarding all pixels falling

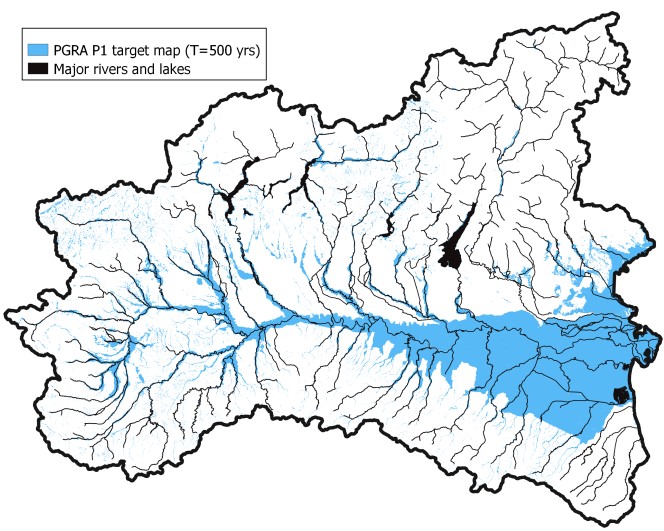

**Figure 3.** Binary flood hazard target map from ISPRA, termed PGRA P1 in this study

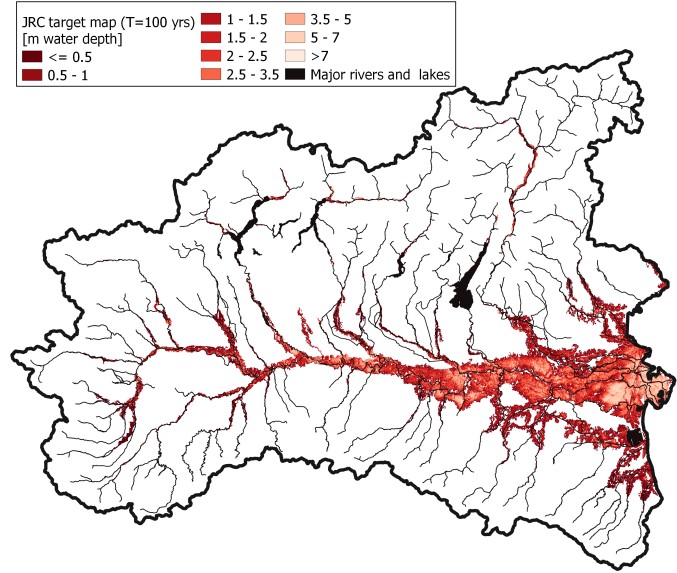

**Figure 4.** Water depth for the target 100-year flood hazard map obtained by Dottori et al. (2016), termed JRC 100 in this study

oustide of an informative calibration area from the dataset (see Sect. 4.1); (ii) splitting the dataset (i.e., pixels falling inside the
190 calibration area) into a training and a test set; (iii) defining the best hyperparameters of DTs through k-fold cross-validation;
(iv) training the DT with hyperparameters from point (iii) and training set from (ii); (v) validation of the DT with the test set.



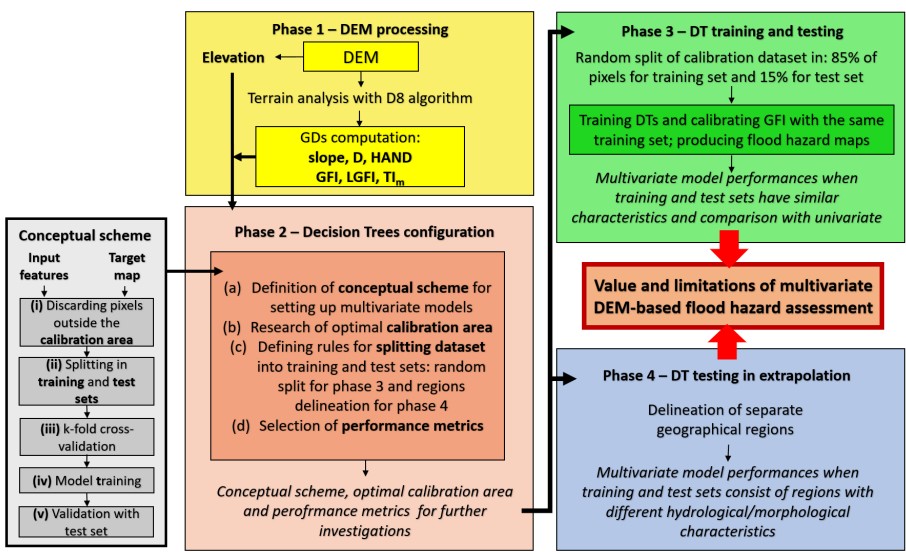

**Figure 5.** Workflow of the analysis

The other steps of phase (2) aim to define (b) the best calibration area, (c) rules for splitting the dataset into training and test set (see Sect. 4.2), and (d) meaningful performance metrics for analysing models' output.

## 4.1 Calibration area

Previous studies (e.g., Tavares da Costa et al., 2019) have highlighted that the DEM-based classification of regions into flood-prone and flood-free zones, is more effective if the calibration is done on meaningful areas. This is due to the different importance of far-from-river and close-to-river pixels in the computation of the objective function. In the present study, training and testing of the models have been performed referring to a portion of the entire study area, which we term calibration area. Different methods to define this area have been tested in step (b) of phase (2), finding that the most effective way of defining 200 a calibration area, representing a good trade-off between the calibration efficiency and the ease of identification, is to refer to a constant-radius buffer around the target flood hazard map. In particular, based on sensitivity analyses that clearly showed that the radius value has a non-negligible impact on the accuracy of the trained model, a 2 km radius has been selected for the PGRA P1 target map, while a 5 km radius has been preferred for the JRC 100 map (see Figure 6). All the pixels falling outside the calibration buffer areas have been neglected when fitting the models, and evaluating the results.

## 4.2 Definition of training and test sets, and geographical extrapolation

All the models considered in this study are trained and tested in different sub-domains of their calibration area. Two different strategies are adopted to divide the input dataset into training and test sets. In phase (3), the pixels of the calibration area have been randomly split in 85% for the training, and 15% for the test set. In this way, the investigation shows the potential of the



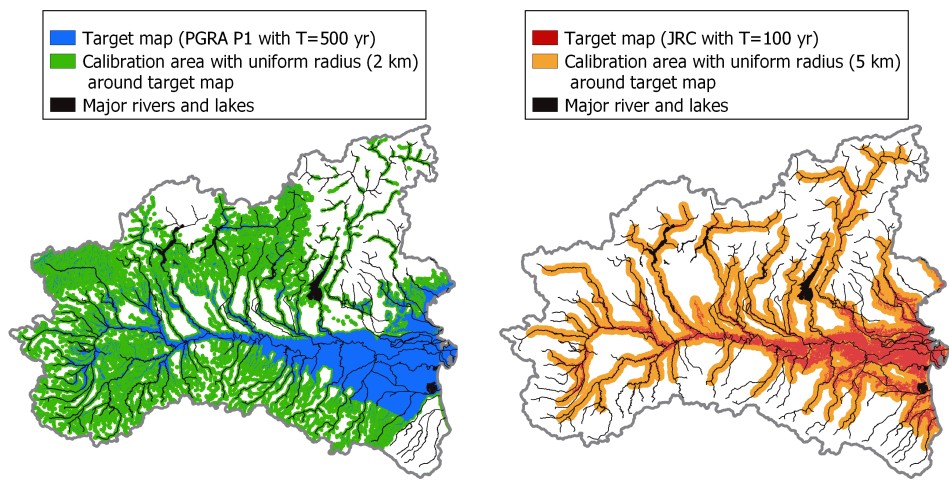

**Figure 6.** Calibration areas: 2km buffer (green) and PGRA P1 flood-prone areas (blue) used for the classification problem (left); 5km buffer (orange) and JRC 100 flood-prone areas (red) used for the regression problem (right)

models in an ideal situation, in which the training is performed with any kind of input and target information, so that analogous

accuracy should be expected for the training and the test sets. In phase (4), four different portions of the overall calbiration area are used for training eigth different decision trees (i.e., four for the classification and four for the regression problem), while the corresponding four remaining portions of the calibration area have served for testing the models themselves. During phase (2), the delineation of these areas, follows catchment boundaries, as well as precise geographical and hydrological criteria (see Figure 7):

– Area A includes the Alpine catchments and the northern portion of the Po river floodplain. The complementary test area includes all the Apennines and the southern part of Po plain

   – Area B includes catchments in the upstream sector of the Po river basin, representing part of the Alps and of the Apennines. The complementary test area includes most of the Po plain, and part of the Alps and Apennines

   – Area C is the complementary of area B

– Area D includes the Apennines, Western and Central Alps and the Southern portion of the Po plain

### 4.3 Cross-validation and model parametrization

Before training DTs in phases (3) and (4), model parametrization is performed through k-fold cross-validation (CV), which is a largely used method of model parametrization and selection. k-fold CV consists in dividing the training set into k folds and then performing two consecutive operations: (1) training of the model using k-1 folds; (2) validation of the model using the

225 remaining fold. These two steps are repeated for k times, for all the combinations of the k folds of the training data (see Figure





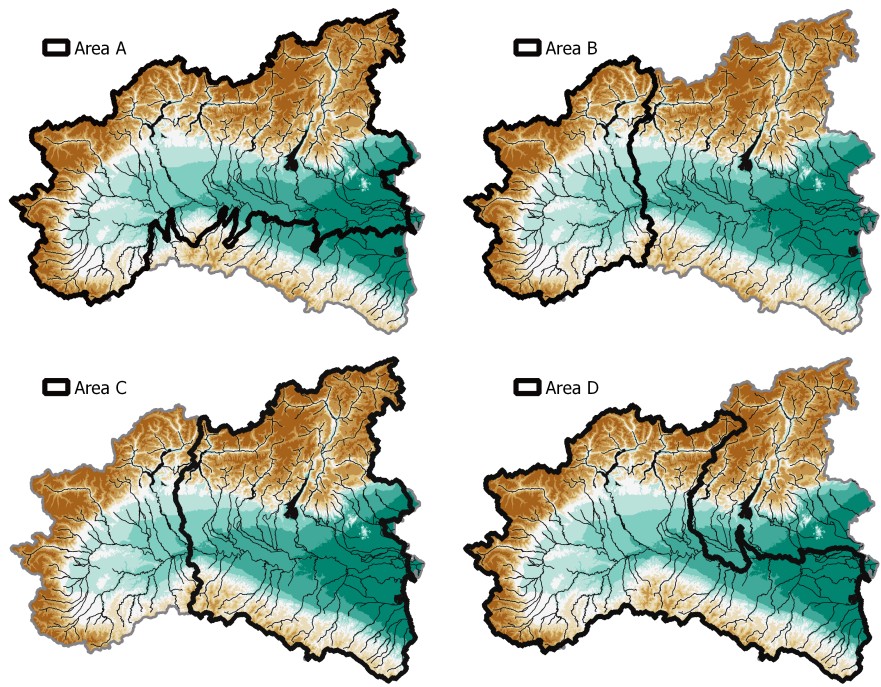

**Figure 7.** Training areas used for the geographical extrapolation experiments performed in phase 4, with major rivers and lakes higlighted in black

8).

k-fold validation is often employed to calibrate the hyper-parameters of the training algorithm. The calibration is performed by picking reference values for each parameter and building a grid with all possible combinations. Then k-fold cross validation is performed to assess the performance of the model trained with the hyper-parameter corresponding to each grid point: the

230 combination of parameters leading to the best performance is then retained. This simple, approximate, optimization approach is sometimes referred to as Grid Search. We use Grid Search with k-fold validation (with k = 5) to optimize the value of two parameters that affect the termination condition of the training algorithm, namely: (1) the maximum tree depth; and (2) the minimum number of examples in any leaf nodes

### 4.4 Objective functions and results evaluation

Specific performance metrics have been used to train the DTs for classification and regression, while other metrics are computed to evaluate their predictions during the validation. With regards to the classification problem, the objective function, used during the training of the DTs, to assess the quality of each split, is the Gini impurity ($I_G(p)$), that varies between 0 (the optimal value) and 1 (Hastie et al., 2009). At each step, the Gini impurity measures how often a randomly chosen element from the set would be incorrectly labeled if it was randomly labeled according to the distribution in the subset. Given the number of target classes




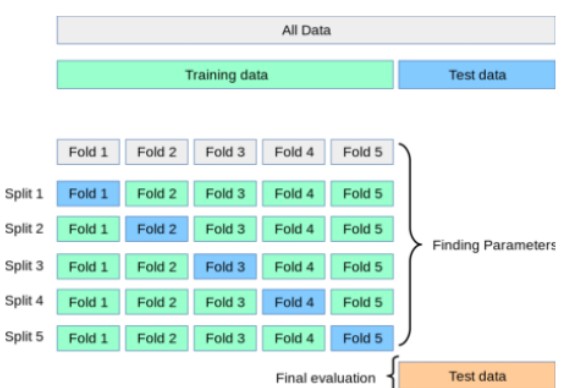

**Figure 8.** Scheme for a 5-fold cross-validation (from: https://scikit-learn.org, see also Pedregosa et al., 2018)

240  J, and the fraction of items labeled with class i in the set $p_i$, the Gini impurity is defined as follows:

$$I_G(p) = \sum_{i=1}^{J} p_i \cdot (1 - p_i) \tag{4}$$

To decide the best parameter set, evaluate and analyze the results in phases (3) and (4), we use the true skill statistic (TSS; Youden, 1950; Everitt et al., 2002):, which is based on the contingency matrix, and varies between 0 and 1 (optimal value):

$$TSS = \frac{TP}{TP+FN} + \frac{TN}{TN+FP} - 1 \tag{5}$$

where TP, TN, FP, FN are respectively true positive, true negative, false positive and false negative predictions of the model. TSS has been successfully used by several authors in different applications (Bartholmes et al., 2009; Alfieri et al., 2012; Tavares da Costa et al., 2019). Some preliminary experiments conducted in step (d) of phase (2) suggested to prefer this metric to accuracy (ACC, see below), which showed to be less sensitive to model modifications (i.e., different calibration areas, input information, tree depth) and goodness (lower extension of FP and FN areas).

Other metrics used for analysing the results are accuracy (ACC), precision (or positive predictive value, PPV), and recall (or true positive ratio, TPR). All the three are very common in evaluating the performance of a classifier (e.g., Manfreda et al., 2015; Samela et al., 2017). They all vary between 0 and 1 (optimal value), and are defined as follows:

$$ACC = \frac{TP+FN}{TP+TN+FP+FN} \tag{6}$$

$$PPV = \frac{TP}{TP+FP} \tag{7}$$



$$TPR = \frac{TP}{TP + FN} \tag{8}$$

With regards to the regression problem, the objective function to minimize during the training is the well-established mean squared error (MSE). Using $n$, $\hat{y}_i$ and and $y_i$ to indicate respectively the number of samples, the predicted and the target value,

MSE can be written as:

$$MSE = \frac{1}{n} \sum_{i=1}^{n} (y_i - \hat{y}_i) \tag{9}$$

The metric used for phases (2), (3) and (4) is the determination coefficient $R^2$, that varies between $-\infty$ and 1 (the optimal value). It measures the improvement of the predicted values relative to the mean of the input samples ($\overline{y}$), defined as:

$$R^2 = 1 - \frac{\sum_{i=1}^{n} (y_i - \hat{y}_i)^2}{\sum_{i=1}^{n} s^n (y_i - \overline{y})^2} \tag{10}$$

Other considered metrics are the above mentioned MSE and the mean absolute error (MAE), defined as:

$$MAE = \frac{1}{n} \sum_{i=1}^{n} |y_i - \hat{y}_i| \tag{11}$$

Lastly, we used the Gini importance ($GI$) to measure the importance of each factor (i.e., each GD) in the trained models (both classifier and regressor DTs), which is defined for the j-th factor as the total decrease in node impurity ($I_{G_i}$), weighted by the fraction of samples reaching that node ($n_i$). Although this measure is largely used for its speed of computation, it has

270 the drawback of neglecting the weakest factor when two related factors are used, which has to be taken into account when discussing the results.

$$GI = \sum_{i=1} \frac{(I_{G_i} - I_{G_{i-1}})}{n_i} \tag{12}$$

## 5 Results

The reliability of the predictions of the models is assessed by performance metrics that refer to (a) the training set and (b) the

275 test set. The distinction between training and test sets is important: while the metrics computed for the training set assess the reliability in reproducing the observed target map, the metrics regarding the test set measure the ability of the model when applied to a different sample than the one used in training (i.e. validation of the model). In order to find out the relevance of each input GD in the DTs' structure, the Gini importance (see Sect. 4.4) for each model is reported in Table 1, and will be better discussed in Section 6.

### 5.1 Delineation of flood-prone areas

Figure 9 represents the flood hazard map obtained with a DT model (i.e. multivariate flood hazard map). To understand the quality of the proposed approach and profitably discuss the results, a flood hazard map has been produced with a univariate




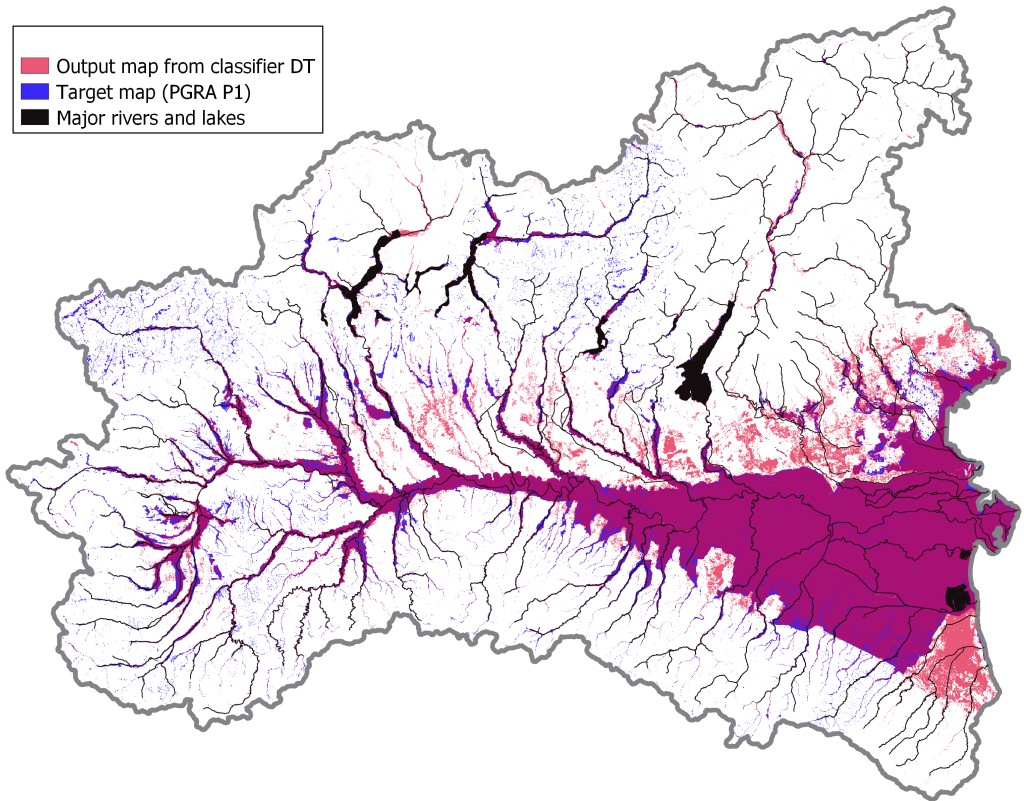

**Figure 9.** Multivariate 500-year flood hazard map for the study area (red), target map (PGRA P1, blue); purple indicates overlaying areas

approach by calibrating a threshold value of GFI in the same training area within the 2 km buffer area used for classifier DTs. Figure 10 illustrates the univariate flood hazard map, while relevant performance metrics for multivariate and univariate models

are reported in rows 1 and 2 of Table 1, respectively.

Figure 9 and Table 1 highlight that the DT flood hazard map is strongly consistent with target map PGRA P1. Also, the model produces a rather detailed mapping across floodplains of minor streams (i.e. exhaustiveness, as defined in Sect. 3); in particular, it can be observed in Figure 9 that the zones where the target map has high exhaustiveness (e.g., northwestern portion of the study area) are mapped with slightly lower exhaustiveness by the DT model, while the DT output is more detailed in floodplain

of minor streams than the target map, where the latter is lacking exhaustiveness (e.g., northeastern part).

Figure 9 shows that GFI uniformly and exhaustively estimates flood hazard along all minor streams in mountain areas, but tends to severely overestimate the size of flood-prone areas in predominantly flat regions.

The first line of Table 3 reports the Gini importance for the classifier DT: HAND scores about 65%, followed by elevation (16.5%) and GFI (10.5%).

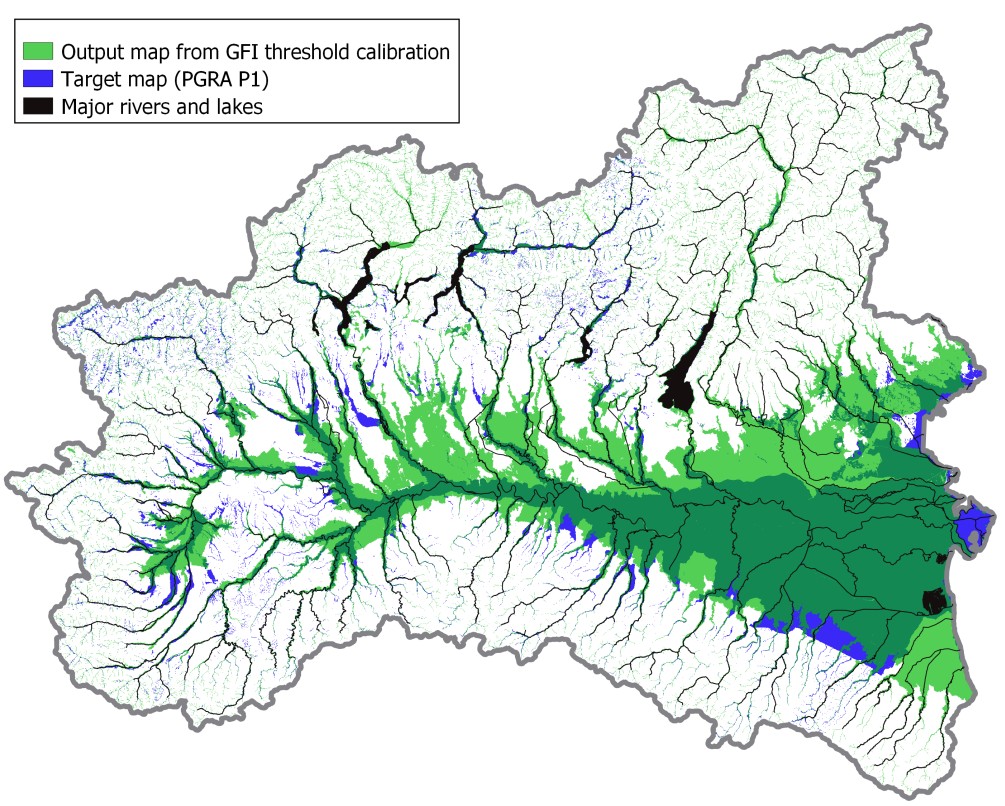

**Figure 10.** Binary flood hazard map resulting from a univariate analysis (morphometric index: GFI, light green); target flood hazard map (PGRA P1, blue); dark green indicates overlaying areas

**Table 1.** Classification problem: performance metrics for the multivariate (classifier DTs) and univariate (classifier GFI) flood hazard maps; target flood hazard map for both approaches: PGRA P1. The reported values have been converted from the interval 0-1 to the percentage notation

| Model | Training area | | | | Test area | | | |
|---|---|---|---|---|---|---|---|---|
| | TSS | ACC | PPV | TPR | TSS | ACC | PPV | TPR |
| Classifier DT | 80% | 93% | 89% | 84% | 78% | 92% | 88% | 83% |
| Classifier GFI | 69% | 84% | 66% | 87% | 69% | 84% | 66% | 87% |
| Classifier DT trained in A | 75% | 92% | 86% | 78% | 56% | 83% | 88% | 61% |
| Classifier DT trained in B | 61% | 93% | 82% | 64% | 65% | 85% | 80% | 75% |
| Classifier DT trained in C | 82% | 92% | 89% | 88% | 33% | 88% | 71% | 35% |
| Classifier DT trained in D | 80% | 94% | 91% | 93% | 63% | 79% | 53% | 87% |



Natural Hazards
and Earth System
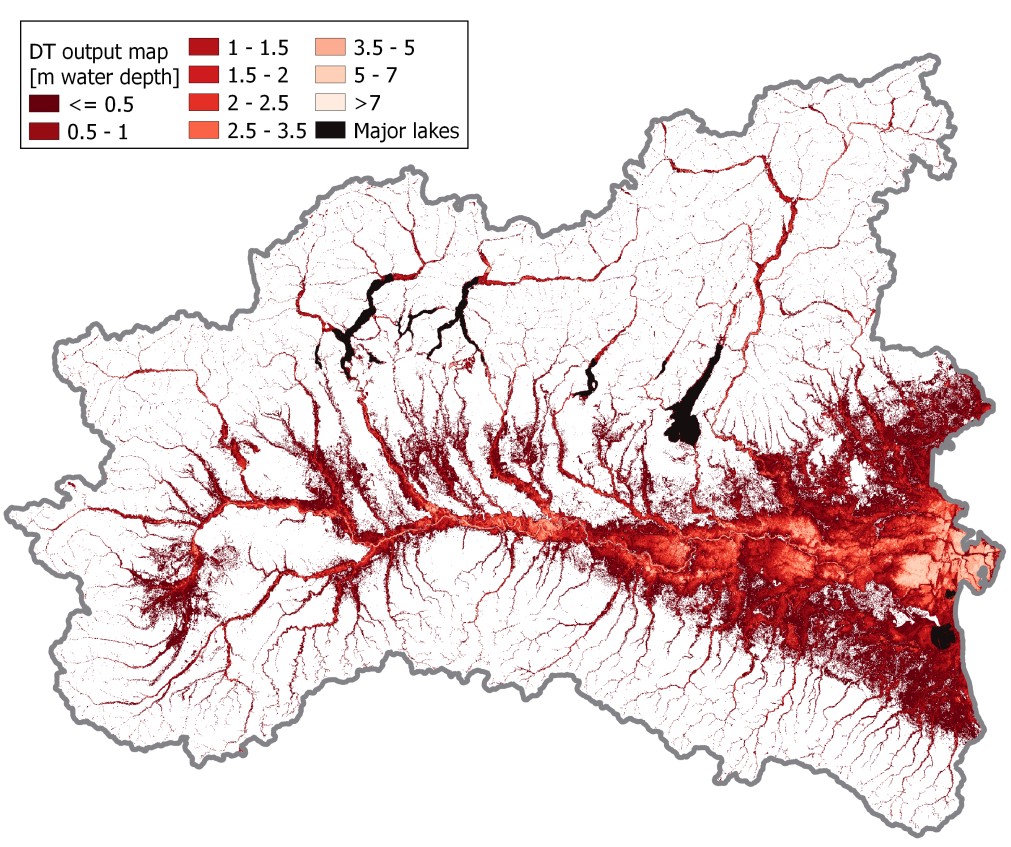

**Figure 11.** Regression problem: multivariate water-depth hazard map obtained with regressor DT (target flood hazard map: JRC 100)

## 5.2 Prediction of flood hazard intensity

Figure 11 illustrates expected maximum inundation water depths as predicted through a regressor DT, relevant performance metrics can be found in the first row of Table 2. Figure 11 and Table 2 show good performance of the DT model for the regression problem. It is worth noting here that the exhaustiveness of the DT water-depth map is considerably higher than that of the reference map (i.e. JRC 100). This result was expected due to the focus of JRC 100 on larger catchments.

The data density plot in Figure 12 depicts the relationship between target and predicted water depths for the test set focusing on true positives (i.e. both target and predicted water depths are higher than 0.0 m) and neglecting water depths higher than $3.5m$, therefore neglecting the very high water depth for both target and predicted values (data pairs beyond axes' limits are 4.2% of the total).

The second row of Table 3 shows that the most informative GD is GFI (63.7%), followed by elevation (20.7%) and sd8 (5.4%).

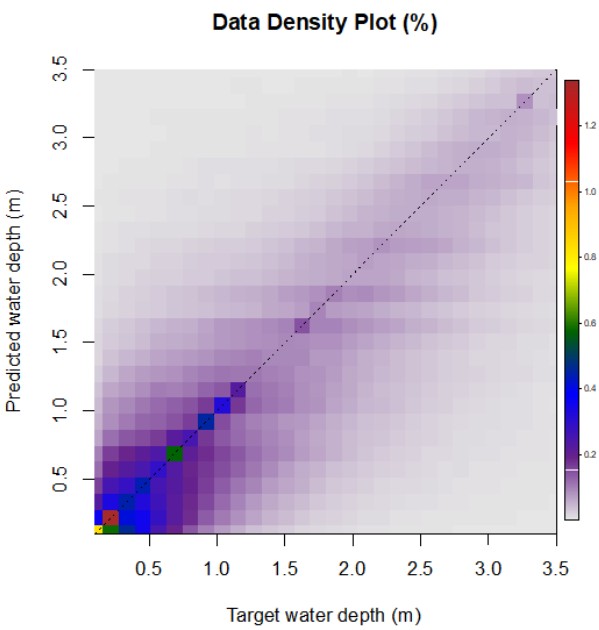

**Figure 12.** Data Density Plot (%) for target vs. predicted expected maximum water depth (target values: empirical JRC 100; predicted values: regressor DT applied to the test set)

**Table 2.** Regression problem: performance metrics for the multivariate water-depth hazard map output maps obtained with the regressor DTs (target flood hazard map: JRC 100)

| Model | Training area | | | Test area | | |
|---|---|---|---|---|---|---|
| | $R^2$ | MSE | MAE | $R^2$ | MSE | MAE |
| Regressor DT | 0.726 | 0.227 | 0.393 | 0.692 | 0.242 | 0.439 |
| Regressor DT trained in A | 0.709 | 0.240 | 0.443 | -0.029 | 1.100 | 0.547 |
| Regressor DT trained in B | 0.606 | 0.145 | 0.284 | -2.110 | 5.208 | 1.283 |
| Regressor DT trained in C | 0.711 | 0.281 | 0.467 | 0.333 | 0.623 | 0.264 |
| Regressor DT trained in D | 0.741 | 0.251 | 0.380 | 0.175 | 1.109 | 0.417 |

## 5.3 Geographical extrapolation of flood hazard assessment: results

Tables 1 (rows 3-6) and 2 (rows 2-5) report performance metrics for the geographical extrapolation experiments for the classification and regression problems, respectively, while Figures 13 and 14 depict the corresponding DT flood hazard maps.

With regards to the classification problem (Table 1), the performance metrics highlight a generalized good agreement with the target map. Figure 13 and section "Training area" of Table 1 show that all models can accurately reproduce the target map in the training area, but they are quite inaccurate in the test area, as it is evident the difference between the two. In fact, concerning


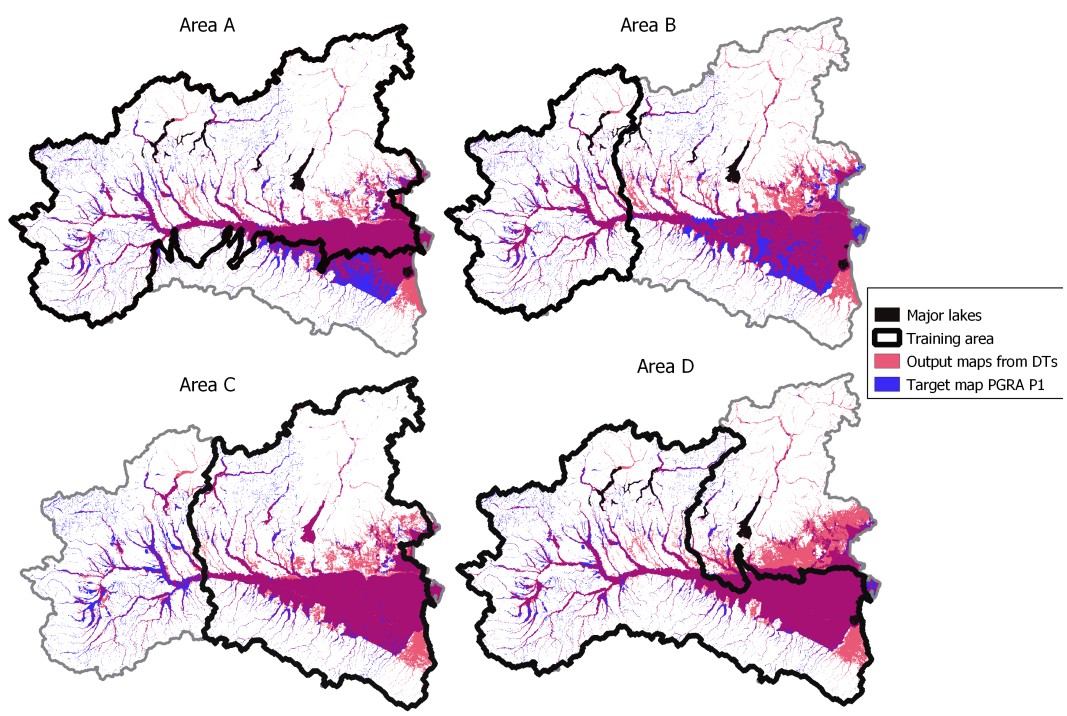

**Figure 13.** Geographical extrapolation for the classification problem: multivariate flood hazard maps obtained from classifier DTs areas (see also Figure 3; target flood hazard map: PGRA P1)

the test area, Table 1 shows that according to the true skill score (TSS), the best prediction in the test area is obtained using B as training area (TSS=65%), followed by D (TSS=63%) and A (TSS=56%), respectively. The same table section shows that the best results are obtained when training on area C if one focuses on accuracy (ACC=88%), followed by B (ACC=85%) and A (ACC=83%). According to precision (PPV), the best result is obtained training the model on area A (PPV=88%), while it is

D according to recall (TPR).

Concerning the regression problem, worse predictive skill in geographical extrapolation is observed in Table 2. Differently from the classification, performance metrics for the regression problem are in good agreement among each other, showing that area C has the better results, while area B is the worst. On the other hand, Figure 14 suggests that water depth estimation in the test area is quite reliable in all the cases, with the exception of the DT trained in area B.

Focusing on Gini importance, Table 3 clearly shows that regressor DTs (rows 7-10) are characterized by similar structures regardless of the training areas: GFI is always ranked first in terms of relevance, followed by elevation and slope. This is not true for the classification problem (rows 3-6): in this case classifiers DTs identified for different training areas have different structures, in which, the most informative geomorphic descriptor can be alternatively GFI, or HAND, or the elevation; this latter is always ranked second.



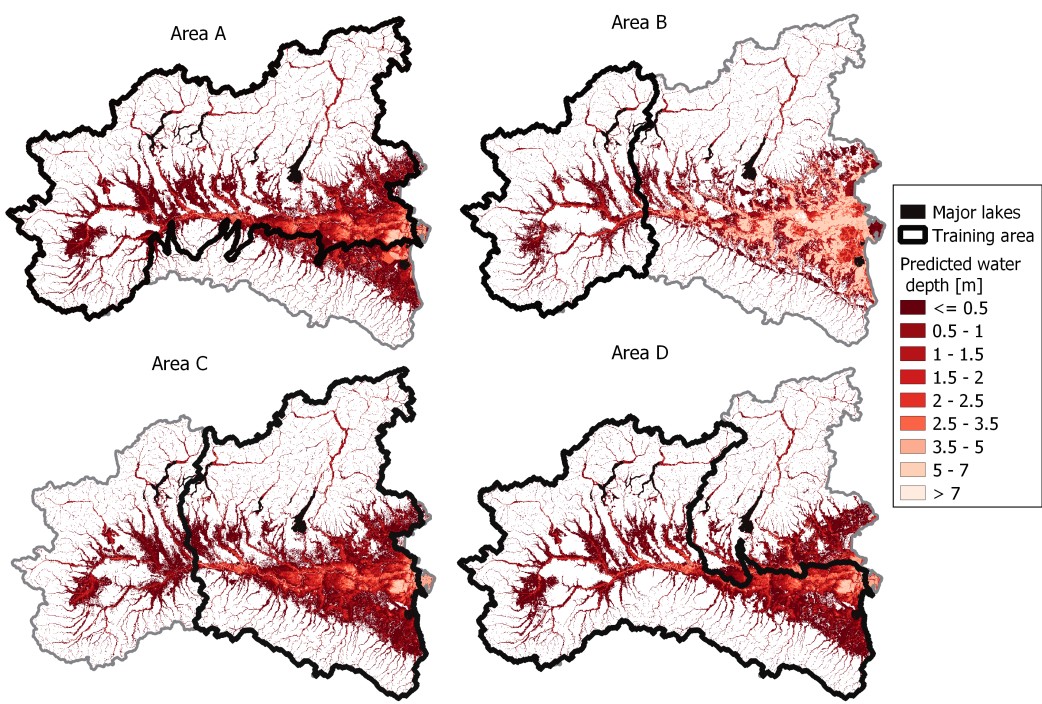

**Figure 14.** Geographical extrapolation for the regression problem: multivariate flood hazard maps obtained from regressor DTs areas (see also Figure 10; target flood hazard map: JRC 100)

**Table 3.** Gini importance of the selected input features computed for the DTs trained in phase 3 and 4

| Model | elevation | sd8 | D | HAND | GFI | LGFI | $TI_m$ |
|---|---|---|---|---|---|---|---|
| Classifier DT | 16.5% | 3.5% | 2.8% | 65.6% | 10.5% | 0.6% | 0.4% |
| Regressor DT | 20.7% | 5.4% | 2.0% | 4.8% | 63.7% | 1.8% | 1.6% |
| Classifier DT trained in A | 10.2% | 6.8% | 2.2% | 8.0% | 71.6% | 0.3% | 0.8% |
| Classifier DT trained in B | 9.8% | 9.8% | 3.8% | 60.0% | 11.8% | 4.2% | 0.4% |
| Classifier DT trained in C | 74.3% | 2.3% | 1.7% | 9.7% | 11.1% | 0.6% | 0.1% |
| Classifier DT trained in D | 18.5% | 2.8% | 1.4% | 69.5% | 7.1% | 0.4% | 0.3% |
| Regressor DT trained in A | 14.3% | 3.6% | 1.8% | 3.5% | 73.2% | 2.3% | 1.3% |
| Regressor DT trained in B | 18.9% | 3.8% | 2.6% | 4.2% | 66.7% | 2.0% | 1.9% |
| Regressor DT trained in C | 17.8% | 3.1% | 1.9% | 4.3% | 69.2% | 2.5% | 1.2% |
| Regressor DT trained in D | 14.3% | 3.9% | 1.3% | 4.0% | 74.7% | 0.9% | 0.9% |




## 6 Discussion

### 6.1 Can we profit from a blend of various geomorphic descriptors for flood hazard assessment and mapping?

The first goal of the present research is the evaluation of the improvement which can be obtained by applying a machine-learning aided multivariate DEM-based flood hazard assessment relative to a univariate DEM-based approach. First, regarding the classification problem (i.e., differentiation between flood-prone and flood-free areas), the outcomes reported in Figures 9-10 and Table 1 (rows 1-2) suggest that the combination of multiple geomorphological descriptors (GDs) increases the comprehensiveness of the morphological description of the study area, and the resulting multivariate data-driven model can reproduce the reference flood hazard map in a significantly enhanced way relative to a univariate approach adopting a single GD. This is particularly visible from the lower extension of wrongly-predicted areas (i.e., false positive, or FP, and false negative, or FN) in the classifier DT output map (ligth red and blue areas in Figure 9) relative to the GFI output map (light green and blue areas in Figure 10).

Second, concerning the regression problem (i.e., prediction of the flood intensity, such as the expected maximum water-depth associated with a given probability of occurrence) the regressor DT considered in our study shows high accuracy in reproducing the target map. Also, it is worth highlighting that regressor DTs provide a direct estimate of this variable, relative to the traditional univariate DEM-based approaches, which usually requires the prior delineation of flood extent to compute water depth, as the elevation difference between the flood-extent border and each pixel (see Manfreda and Samela, 2019). Figure 12 highlights that the correlation between the predicted and target water depths can be improved, yet it also clearly shows that predictions for the test set are unbiased. It is worth mentioning here that the diagram neglects the true negatives (i.e. target and predicted water depths are equal to $0.0m$; 49.78% of the cases), false positives (i.e. only predicted water depths are equal to $0.0m$; 22.37% of the cases) and false negatives (i.e. only target water depth are equal to $0.0m$; 0.08%). While the occurrence of the most concerning cases (false negatives) is very limited, predictions show significant margins for improvement as far as the false positives are concerned. Nevertheless, it should also be recalled here that the target map by its own very nature neglects smaller streams (contributing area has to be higher than $500km^2$), whereas the decision tree regressor looks at morphology only and provides water depth predictions also for smaller streams (i.e. higher exhaustiveness, see Figure 12).

One of the most interesting aspects is the relevance that each GD assumes in the regressor DTs (see Table 3). It can be observed that all models rely mainly on one single GD, with Gini importance always in excess of 60%, but still, the multivariate analysis leads to significantly better results relative to the univariate one. Also, it is important to highlight that:

– While regressor DTs tend to depend mainly on the GFI, classifier DTs depend on HAND

– While the input GDs have quite a similar Gini importance hierarchy in regressor DTs, classifiers DTs assume different structures in the considered cases

– All models agree in giving low Gini importance to LGFI and $TI_m$

– Elevation is very often ranked second, always showing significant importance





Overall, this suggests that DT regressors tend to operate by correcting a baseline estimate that mostly relies on the GFI value. On the other hand, DT classifiers obtain their results by following different rules depending on the training data, and often prefer using lower-levels features relative to more complex indicators such as the GFI. It should be kept in mind, however, that different Gini importances do not necessarily imply radically different classification rules, due to the existing correlations between the input features. Ideally, dedicated feature selection and importance analysis algorithms should be used to obtain deeper insight on how the different models come to their conclusions; we plan to investigate this line as part of future work.

### 6.2 Can we use simple ML techniques for effectively blending multiple GDs?

The second research question of the present study is wheter it is possible to obtain a good estimation of flood hazard by combining multiple GDs with low complexity machine learning models. Differently from several other works in the literature, we do not focus on model complexity or on the comparison of different models (Wang et al., 2015; Khosravi et al., 2018; Mosavi et al., 2018; Arabameri et al., 2019; Costache et al., 2020). Instead, we prefer to select one simple model type (i.e., decision trees, DTs) and focus on the combination of the five innovative elements listed in the Introduction; in this way, we can analyse the influence on the multivariate DEM-based approach of the preliminary steps, consisting in data pre-processing (i.e., selection and manipulation of input features, target maps, training set and test set). This is highly important, because machine learning models do not reproduce the dynamics of the water, as such, their performance is strictly linked to the data used for the training, that need to be handled very carefully.

As it is highlighted in Sec. 6.1, the outcomes of the study (Figures 10-11, Tables 1-2) clearly show that DTs can effectively reproduce the target information (Figures 3-4) with high accuracy for both classification and regression problem, even if the resolution of the MERIT DEM (Yamazaki et al., 2017), from which the input GDs have been retrieved, is not very high. Moreover, it is worth mentioning that the trained DTs estimate flood hazard associated with different minor streams that are neglected in the target maps (see red areas in Figure 9; compare Figure 11 with Figure 4): due to the absence of information in these areas, it is not possible to assess the goodness of the models output, but this tendency of completing target information could be a key aspect for future applications to data-scarce regions, and thus, it could be considered as a promising characteristic of the models.

Overall, it is possible to observe that DTs are effective tools to combine geomorphic descriptors and estimate flood hazard. This indicates that proper data handling has a strong influence on the accuracy of the final estimation, which is comparable to the choice of a given machine learning technique. In particular, we want to underline two elements of the presented approach that have great importance on the predictive skill. First, the utilization of flood hazard maps as target results in a large number of pixels for the training and test set, and therefore a very broad spectrum of hydrological/morphological characteristics, which represent a much more informative dataset relative to isolated points used by other authors for training more complex models (Lee et al., 2017; Khosravi et al., 2018; Arabameri et al., 2019; Janizadeh et al., 2019). Second, a sensible identification of a calibration area is very important for a successful training, as it allows to neglect irrelevant pixels. To this aim, a preliminary sensitivity analysis might be very useful for identifying the optimal buffering radius around the target map (see Sect. 4.1).



### 6.3 Are these techniques capable of providing a reliable assessment of flood hazard over large areas in extrapolation?

The evaluation of prediction accuracy for geographical extrapolation is a key and characteristic aspect of our study. On the one hand, performing predictions with new input data is a major problem for machine learning models (DTs in this case); on the other hand, reaching good predictive skills in geographical extrapolation (i.e. applying classifier or regressor DTs in geographical areas, or watersheds, that have not been considered for parameterizing and training the models) is needed for future practical applications to data-scarce environments. What is more interesting about this part is to understand the link between training and test performances: if the relationship between input and target values, learnt by the model during the training, is also valid for the extrapolation region, accurate test predictions are obtained, but this depends strongly on the choice of input and target datasets for the training, which can be very difficult. Before addressing this very issue, a careful discussion of the resulting metrics and maps is required, as their interpretation is not straightforward.

With reference to the classification problem, each metric suggests a different training area as the best case, and this highlights how difficult it is to choose a single metric for describing the goodness of a model for a binary classification. Figure 13 and TSS values in rows 4-5 of Table 2, could suggest that Area B (test TSS=65%) has better extrapolation performance than Area C (test TSS=33%). On the contrary, ACC is similar for the two cases, and higher for Area C (ACC=88%) than for Area B (ACC=85%), suggesting that TSS is a more informative metrics than ACC in representing the model performance. On the other hand, precision and recall appear to be quite unbalanced metrics, as areas A and D lead to test prediction with considerable overextension of FN and FP values, respectively (see Figure 13). Differently, regression metrics agree in pointing at the DT trained in B as the best case (Table 3). However, the absolute values of $R^2$, that depicts low-accuracy test predictions, do not reflect other metrics (MSE and MAE) and the output maps (Figure 14).

As expected, the choice of the training area has great influence on prediction accuracy. This is particularly visible for the classification problem: in Figure 14, the difference between metrics for training and test is striking. Nevertheless, this difference becomes less clear for the regression problem (Figure 14). The same observations are confirmed by Table 3, where evidence is given of different structures for the classifiers DTs, while the regressor DTs are all very similar. More in detail, the obtained results show that the extent of the training area has less importance than the quality of the input data that it contains. Perfect examples of this observation are classifiers DTs trained in A and D: even if both A and D are very wide, prediction over the test area is affected by considerable errors. This happens because A does not include any part of the Apennines, while D ignores a large flat area in the eastearn coast, meaning that any geographical system corresponds to a specific relationship between input GDs and flood susceptibility, and thus it cannot be fully represented by a model trained with very different datasets. The comparison between area B and C is also meaningful: while the training in B leads to good test predictions for the classification, it is the worst case for the regression (the opposite is valid for C). This is probably due to the fact that area B contains useful information to delineate flood-prone areas, as it represents the upstream section of Po river, but cannot adequately train a regressor DT, as it lacks high target values (i.e., high inundation water depths). To sum up, GDs combination with DTs is capable to provide quite a reliable estimation of flood hazard (i.e., flood-prone areas and maximum water depth)





in extrapolation mode, but a careful choice of the training area is needed, where target and input dataset is complete and representative for the test area.

## 7  Conclusions and further steps

Our study analyses and compares data-driven and resource-efficient methods for assessing and mapping riverine flood hazard across large geographical areas. It illustrates the potential and limitations of combining different geomorphic descriptors by means of decision trees for delineating flood prone areas and for predicting the expected maximum water depths for a given return period. We focus on a large study area in Northern Italy (size $\sim 10^5 km^2$) containing Western, Central and part of the Eastern Italian Alps, part of the Northern Apennines and the floodplains of a complex river-system including the main rivers Po, Adige, Brenta, Bacchiglione and Reno. The morphology of the study area is described by the Multi-Error-Remover Improved-Terrain model (MERIT DEM; see Yamazaki et al., 2017), with a 90-meter resolution, approximately. Decision trees are trained using as input features the geomorphic descriptors retrieved from the MERIT DEM, and as target maps two different datasets: one representing flood extent with a reference return period of 500 years, and one representing expected maximum water depth for a 100-year return period scenario.

Relative to previous studies focusing on morphometric floodplain delineation and flood-hazard mapping (see e.g., Dodov and Foufoula-Georgiou , 2006; Nardi et al., 2006; Manfreda et al., 2011, 2014, 2015; Samela et al., 2017; De Risi et al., 2018) and machine-learning aided multivariate flood hazard mapping (see e.g., Gnecco et al., 2017; Arabameri et al., 2019; Janizadeh et al., 2019; Costache et al., 2020), our study is the first one of its kind that simultaneously combines the following five elements: (a) only strictly DEM-based morphometric data and indices are used for predicting flood hazard; (b) morphological characterization of flood hazard associated with a given probability of occurrence is studied separately as a classification problem (i.e., generation of binary flood hazard maps) and as a regression problem (i.e., prediction of expected maximum inundation water depth); (c) machine learning models (i.e., decision trees) are trained using pre-exhisting flood hazard maps as target information; (d) univariate geomorphological assessment of flood hazard (i.e., one geomorphic descriptor used as predictor) is thoroughly compared with a multivariate assessment, in which several DEM-based geomorphic descriptors are blended together by means of decision trees; (e) potential and accuracy of DEM-based flood hazard prediction is assessed in geographical extrapolation by applying models trained on specific geographical areas to different areas having diverse morphologic and/or hydrological features.

In particular, we address three main science questions: (1) can we profit from a blend of geomorphic descriptors to perform flood hazard mapping with respect to a univariate DEM-based approach? (2) Are decision trees a valid tool for combining multiple geomorphic descriptors? (3) Is this approach capable to predict flood hazard over large areas in geographical extrapolation? With reference to the first and second questions, delineation of flood-prone areas (i.e. binary flood hazard mapping) is derived with two methods: a univariate approach, consisting in the calibration of a threshold value for a given DEM-based morphometric index (i.e., Geomorphic Flood Index, GFI; see e.g., Samela et al., 2017), and the proposed decision tree for multivariate DEM-based classification. Also, prediction of the maximum inundation water depth associated with a 100-year



return period has been carried out. As done in other studies (Tavares da Costa et al., 2019), buffer areas around the target flood-prone areas are defined, in order to discard pixels far from the main river network: the models are trained and tested with different sets, consisting respectively in randomly-selected 85% and 15% of the pixels contained in the buffer. The results obtained for the classification problem show high performance metrics in validation (overall true skill statistic TSS$\sim$80%, overall accuracy ACC$\sim$92%) relative to the univariate approach (overall TSS=69%, overall ACC=83%). In particular, the combination of DEM-based descriptors leads to much more accurate results in flood-prone areas delineation over predominantly flat regions. Concerning the regression problem, good performances are confirmed in validation as well (i.e. overall determination coefficient $R^2 \sim$0.7, overall mean absolute error MAE$\sim$0.4 m).

With reference to the third question, we delineate four different subregions of the study area to train classifier and regressor decision trees by selecting four areas belonging to four different hydrologically-coherent geographical systems. When tested on the reminders of the study area, the four different models show different extrapolation performances depending on the morphological features (e.g. Apennines vs. Alps) and the broadness of the hydrological conditions included in the training subregions. In particular, concerning the classification problem, it is possible to observe that models trained in areas containing headwater catchments of the main rivers can extrapolate better over the downsteam portions of the basins than vice versa. Concerning the regression problem, the selection of the training area must rely not just on these morphological and hydrological features, but also on the availability of a suffcently wide range of values for the target variable (i.e., maximum water depth in our case) within this area, in order to adequately train the model. This means that training in headwater catchment areas performs very poorly for extrapolating maximum water depth across downstream floodplains.

In general, we observe that multivariate DEM-based analysis by means of decision trees is very effective in estimating flood hazard relative to univariate approach, and that these techniques have good potential in extrapolation mode as well. Moreover, output of multivariate DEM-based flood hazard assessment studies may represent a very useful complement to existing large scale flood hazard maps for two reasons: (1) they omogenize mapping when the existing maps have different levels of detail in different regions (e.g., in situations in which the large scale map consists of the merger of maps from different local authorities, which applied different flood hazard assessment criteria and methods); (2) they contribute to assessing the hazard level also in areas not inlcuded in the original mapping (e.g., when smaller river catchments have been neglected).

Different elements of our work can be further examined in future studies, in order to allow better performances of the DEM-based multivariate techniques. First, finer resolution DEMs could be used, in order to increase the accuracy of the morphological description of the study area. Second, to further enhance the input information, soil and climate data (e.g., permeability and precipitation) could be added beside geomorphic descriptors. Finally, more complex machine learning models should be tested, for better characterizing the impact of selecting a given technique on the accuracy of flood hazard assessment.

*Author contributions.* A.M. designed and performed the experiments, wrote the codes and derived the models; M.L. had a key role in the application of machine learning techniques and the definition of the methodology. A.C. supervised and conceptualized the project. S.P.



contributed in supervising the project and solving technical problems with the experiments. F.L.C. and A.T. took part in analysing and discussing the results. A.M. wrote the first version of the manuscript, and all the authors helped in writing the final one.

*Competing interests.*  The authors declare that they have no conflict of interest.

*Acknowledgements.*  The authors would like to thankfully acknowledge Leithà S.r.l. - Unipol Group (regional research grant: "Stima della pericolosità idraulica del territorio italiano") and Autorità di bacino distrettuale del fiume Po (regional research grant: "Idrologia di piena nel distretto del Po") for their financial support and access to data. The authors gratefully aknowledge the use of Free and Open Source Software, in particular Python (Van Rossum et al., 1995), Scikit-learn (Pedregosa et al., 2011), QGIS (QGIS Development Team, 2021), GRASS GIS
(GRASS Development Team, 2019) and TauDEM (Tarboton, 2003).



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
