# Peer review of "Machine-Learning blends of geomorphic descriptors: value and limitations for flood hazard assessment across large floodplains"

_Natural Hazards and Earth System Sciences, 2021_

## Author Comment (AC1)

**REPLY TO REVIEWER SHUANG-HUA YANG**

Dear Dr. Shuang-Hua Yang,

We would like to thank you very much for your comments and suggestions. Your help is strongly appreciated, and we believe it will significantly contribute to the improvement of our manuscript.

We propose to address your comments after dividing them into two major points. For easing the reading of our rebuttal, original comments are reported in italics after the tag "***Reviewer:***", while our reply is flagged using the tag "**Authors**".

1. *Reviewer: The presentation of this manuscript is very clear. The authors provide six geomorphic descriptors to build decision tree model for classification and regression. By nature the DT is designed for calssification rather than regression. The authors need to give more details to describe how the DT can be used for regression.*

   **Authors:** we are very pleased to receive this overall positive evaluation of our manuscript, as well as to answer to this very important point. DTs are commonly used to address both classification and regression problems (see Hastie et al., 2009, https://doi.org/10.1007/978-0-387-84858-7), but indeed, their output always consists of a classification of the input data. This means that even if regression is performed, the continuous range of output values is binned into classes, and so the output of a regressor DT is a discrete estimation of the output variable. The number and distribution of the output classes for the predicted water depth depends on the tree structure (i.e., on the number of nodes and on the configuration), and it is automatically handled by the considered algorithm (i.e., function from software *scikit-learn*, see the manual on the official website https://scikit-learn.org/0.21/_downloads/scikit-learn-docs.pdf) once the tree parameters are decided by the user (maximum depth, minimum samples per leaf).
   In our case, the discretization of the output variable (i.e., water depth) is not a problem. In fact, as we use continuous flood hazard maps as target data and we have millions of observations (i.e., pixels), we can train very complex DTs, with many output classes for water depth. To give an idea of the grade of approximation, the regressor DT that produces the output map of figure 11 has about $10^4$ output classes.
   We will address these comments and make our description clearer in our revised manuscript.

2. *Reviewer: Figure 4 illustrates the water depth has been classifed into different groups <0.5, 0.5-1.0, 1.0-1.5 ...,>7. The reviewer guesses the regression was based on these grouping categories. More information is required*

   **Authors:** Thank you for pointing out this issue, which can lead to a misunderstanding of our results. As we stated in our reply to the Reviewer's previous comment, the classification of our regressor DTs is based on thousands of classes. The legend in Figure 4 (which is the same of figures 11 and 14) explains a classification of the raster values that has been chosen for an effective representation of the maps. We will clarify this aspect in the revised manuscript.

We hope that our letter successfully replies to your comments, that we will address while revising our manuscript.

Thank you again for your appreciated help,

Kind regards

Andrea Magnini, Michele Lombardi, Simone Persiano, Antonio Tirri, Francesco Lo Conti, Attilio Castellarin

---

## Author Response (AR1)

**AUTHORS' REPLY TO THE COMMENTS**

Dear Dr.s Lili Yang, Caterina Samela, Zhejun Huang, Shuang-Hua Yang and Heidi Kreibich,

We would like to thank you all very much. We are grateful for your editing, comments and suggestions. Your help is strongly appreciated, and we believe it will significantly contribute to the improvement of our manuscript.

In the following points, we report our reply to your major comments and our actions to address them:

1. ***Reviewer 1 (CS):*** *Terminology: "flood hazard" maps is a terminology more appropriate to maps derived by hydrologic/hydraulic simulations. Topography-based (hydrogeomorphic) maps are generally termed in literature as flood-susceptibility maps, or flood-prone areas map or floodplain maps (see e.g. indersson, S., Brandimarte, L., Mård, J., and Di Baldassarre, G.: Global riverine flood risk – how do hydrogeomorphic floodplain maps compare to flood hazard maps?, Nat. Hazards Earth Syst. Sci., 21, 2921–2948, https://doi.org/10.5194/nhess-21-2921-2021, 2021.)*

   **Authors:** Thank you for pointing out our misleading terminology. We will revise our manuscript thoroughly according to your useful suggestion.

   **Actions:** We have adjusted the terminology used when referring to the produced maps, changing them with the term "susceptibility maps".

2. ***Reviewer 1 (CS):*** *One of the most important issues addressed in this work is the estimate of the water depth, a parameter of fundamental importance especially for estimating expected flood damage. Compared to the large number of published studies on the delineation of the areal extent of flood hazard areas, in the literature there are fewer studies concerning the estimation of water inundation depth with simplified methods. This is an added value of this work. However, since DEM-based methods find their primary purpose in applications in data-scarce environments (although not exclusively), it is perhaps worth because while reference data to calibrate the classification problem are often available also in these contexts, on the opposite flood hazard map providing water depth values (to use for calibrating the regression problem) are more difficult to find. In addition, this data should be characterized by good accuracy to train a simpler but reliable model based on it. I think a consideration on this aspect can find a place in the manuscript.*

   **Authors:** This observation focuses on a particularly important aspect of our work that we did not address in detail. Indeed, predicting the expected maximum water depth is more complex than delineating flood-prone areas, and this requires more accurate data to effectively train machine learning algorithms. A possible solution to the application of our approach to data-scarce environments is to use the outcome of continental/global studies, when they are available (see e.g. the dataset provided by the Joint Research Centre at 25 m resolution at global scale; this dataset, that is the one we used to train our models, should be sufficiently reliable to train and

validate machine learning algorithms, even if it does not consider –as in our case study- watersheds with lower extension than 500 km$^2$), or the outcomes of detailed hydraulic studies in specific portions of the study region, again, where and when available. As suggested by the reviewer, we will include some considerations on this truly relevant issue, by revising the introduction and discussion sections.

**Actions:** We added a comment on this part at lines 402-404.

3. ***Reviewer 1 (CS):*** *I wonder about the choice of identifying the calibration area by setting a constant-radius buffer. In this study, testing the performance outside of calibration areas is part of the application, so it was possible for authors to perform a sensitivity analysis on the accuracy obtained with different buffers. However, readers who want to apply the methodology with no possibility to validate the results (e.g. in poor data environments) are left without guidance on how to set this constant buffer. Here, in the same work for the same study area, two different buffer values are considered the best for the two reference maps (2 km for the 500-years PGRA flood hazard map, and 5 km for the JRC 100-years flood map). Instead, a topographical-hydrological criterion (e.g. the one used by Degiorgis et al., 2012) offers the possibility of being adopted and re-applied in any context, responding at the same time to the characteristics of the study area and of the available reference map. This consideration does not influence the relevance of the investigation and the interesting results obtained, but is made thinking about how to replicate the study in different case studies.*

**Authors:** The point highlighted in this comment is truly relevant, and we are pleased to have the occasion to elaborate further on it, while revising our manuscript. During our research, several experiments have been performed trying different calibration areas, that can be divided into two groups: the first consists of the merger of all the elementary basins (i.e. hydrological units that drain directly into an elementary stretch of the river-network) that are entirely or partially included in the target flood-prone areas; the second consists of the areas within a fixed buffer around the target flood-prone areas. Testing these alternatives, i.e. training our classification models on these two alternative calibration areas, we observed the same performances of the trained models, as opposed to an enormous difference in terms of computational effort for defining the calibration areas associated with the two techniques (i.e., buffering the target map is computationally more effective). For this reason, further analysis has been conducted with a fixed buffer calibration area.

The same approach to define a specific area to calibrate the models has been proposed and used by Tavares Da Costa et al. (2019), who used a fixed buffer of about 1 km around the flood-prone areas of the target map, and named this as "classification area". The approach by Degiorgis et al. (2012) is remarkably similar, as they considered for the training just the pixels within flood-prone target areas and their conterminous.

During our research, we observed that the choice of the radius for the calibration buffer has some influence on the results, and, it needs to be large enough to enable the model to recognize non-flood-susceptible pixels. Also, as stated in the manuscript, different radius can perform differently depending on the target map being considered.

Our manuscript does not provide the interested reader with enough detail on the choice of the

right buffer radius for the calibration area, which requires some sensitivity analysis. We will better detail this part and suggest to resort to Tavares da Costa et al. (2019) or Degiorgis et al. (2012) when a sensitivity analysis is not viable (i.e. possibility to validate the model).

**Actions:** we inserted a comment about this topic in lines 400-402.

4. *Reviewer 1 (CS): The analyses are made up of a series of steps and sub-steps (and further sub-steps), not always easy to follow along, that are listed in the first lines of Section 4 "Framework of the analysis". Then, subsections 4.x do not follow any of the previous subdivision. Did you consider that the methodology would be easier to read, follow and reproduce if a subsection is dedicated to each of the major 4 steps?*

**Authors:** We appreciate this suggestion, as it highlights a key section of the manuscript that can be significantly improved. We agree with this comment, and we think that the structure of Section 4 should be redesigned to be more linear and representative of our methodology.

**Actions:** as we totally agree about the difficult interpretation of the first version of Framework Section in our manuscript, we totally redesigned it. It has now a different and clearer structure, no framework figure and no cross-validation figure. We hope that now our work is easier to understand and read (see Section 4, Framework of the analysis).

5. *Reviewer 1 (CS): In tables 1, 2, 3 can be unclear the difference among the results of the first two rows and the other rows. Section 4.2 reports that the models have been applied a first time using the entire domain of the calibration areas, and then the models were applied again four other times after selecting four subdomains of the calibration area (to test extrapolation performances). I believe this should be better clarified and the section 4.2, in general, could be reorganized. For example, it first describes what happens in phase (3), then in phase (4), and toward the end of the section is nominated phase (2). Is there a possibility to simplify and re-order this description?*

**Authors:** Many thanks, the difference of what we did in phase (3) and phase (4) is a fundamental aspect of our study. In phase (3) all the pixels in the calibration area were randomly divided into two groups: 85% for to train the models, and 15% for the validation. As the split was random, the datasets for the training and the validation have the same statistical distribution of the different values for the seven input indexes and for the target values. In phase (4), the pixels in the calibration area have been divided into training and validation sets with a geographical meaning. This leads to applications of the same approach to real world applications, where the input dataset has some hydrological and morphological characteristics depending on its morphological and hydrological features, and the validation set has different characteristics. We are extremely glad that this has been pointed out; we will rewrite Section 4 in a more effective and clear way.

**Actions:** as we described in the previous point, the structure of Section 4 has been redesigned, and we hope this makes the overall manuscript more intelligible.

6. *Reviewer 2 (ZH): In the present study, an approach that blends several geomorphic descriptors together using the Decision tree method is proposed and is compared with a univariate approach, Geomorphic Flood Index (GFI). Only DEM-based geomorphic descriptors from pre-existing flood hazard maps were used. The decision tree method was employed to establish two types of models: one for classifying flood-prone areas and one for predicting water depth. Seven distinct geomorphic descriptors (GDs) were considered and blended in the multivariate approach. They demonstrated that the performance of the proposed multivariate approach was better than the univariate approach using GFI only.*
   *The article is well written and easy to read. However, tables need to be improved to be more readable.*

   **Authors:** we thankfully acknowledge the overall positive evaluation of our manuscript, and take advantage of this further opportunity to thank the Reviewer for his useful comments. We will improve the readability of all tables.

   **Actions:** Tables in the new version of the manuscript have been changed. Lines in tables 1 and 2 have alternate gray/white colors. Notable metrics have been reported in bold and italic, and legends have been improved in all the tables (see Tables 1, 2, 3).

7. *Reviewer 2 (ZH): In the present study, a new approach that blends several GDs together is proposed. This work indeed fills some gaps in the field of flood hazard assessment, the techniques and methods are feasible. I consider this work a contribution in this field and this manuscript can be accepted for publication after revisions. Only one univariate using the geomorphic descriptor GFI is used as the comparison method. Is the performance of the univariate approach using GFI better than the other ones using the other six GDs?*

   **Authors:** Thank you for focusing on this point, that is very important for the evaluation of the results. We compared the multivariate approach with the GFI-univariate only based on: (1) several contributes in the literature indicating GFI as one of the most informative morphometric indexes for floodplain delineations (e.g., Samela et al., 2017, https://doi.org/10.1016/j.advwatres.2017.01.007), and (2) our preliminary analyses, all showing that GFI has better performances than any other index considered in our study. We will explain with more detail this point in the next version of the manuscript.

   **Actions:** a clarification of this point has been added in the new Section 4, in lines 197-198.

8. *Reviewer 2 (ZH): Does the proposed approach have advantages over the existing multivariate approaches? I would suggest comparing the proposed multivariate approach with one or two existing multivariate approaches to make this study more comprehensive and convincing.*

   **Authors:** The point highlighted in this comment is relevant, and we are pleased to have the occasion to elaborate further on it, while revising our manuscript. Our study does not aim specifically to propose a more accurate multivariate approach. We focused on an alternative way to approach multivariate DEM-based flood hazard assessment that differs significantly from other multivariate approaches presented in the literature (see e.g., Janizadeh et al, 2019,

https://doi.org/10.3390/su11195426; Costache et al., 2020, https://doi.org/
10.1016/j.jenvman.2020.110485); we tried to summarize the innovative aspects in the
introduction (mostly) and conclusion sections of the manuscript, and we will revise these
sections to make our message as clear as possible.

Our approach is associated with two main advantages relative to existing methods. The first
advantage is the feasibility and repeatability, as we only used descriptors that can be easily
retrieved from DEM processing, while other studies exploited additional information (e.g., about
geology, soil type, precipitation) that needs more extensive research and could be in some cases
unavailable or unreliable. The second main advantage is the reliability, as our study does not use
as target information records of historical events, instead, our model is trained and tested
against previously published flood hazard maps, which consist of ensembles of hydraulic model
output and represent scenarios with a certain return period.

Due to the second point, it is quite difficult to compare performance metrics of our model with
the ones of other studies due to the intrinsic differences among them; also, our performance
metrics are computed pixel-based on a continuous domain, while other multivariate flood
hazard assessments simply refer to sparse geographical locations that were inundated or not by
specific historical flood events.

Undoubtfully, the comparison between our approach and other multivariate models is very
interesting. Nevertheless, our aim is not to improve the accuracy of multivariate DEM-based
approaches, instead, we want to evaluate the benefits derived from the combination of
different DEM-based descriptors compared to a univariate approach. Indeed, the Reviewer's
suggestion is a good topic for further studies.

**Actions:** A new suggestion for further studies has been added about this point in the Conclusion
(lines 496-498).

9. ***Reviewer 2 (ZH):*** *How to determine the ratio of training and testing set and why? The authors
should clarify it since the ratio is quite crucial for the learning model and thus affects the
performance of the approaches. Usually, the performance in the test set is accepted rather than
that in the training set. However, the manuscript used the results in the training area in the
abstract (as shown in Table 1).*

**Authors:** We sincerely appreciate this comment, as it allows us to give more details about some
important aspects of the study. The ratio between training and testing followed two different
rules during two consecutive phases of the study. First, we divided the entire dataset (i.e., the
calibration area, consisting of the floodable zone close to rivers with buffer) into 85% for
training and 15% for testing, based on established proportion adopted for machine learning
algorithms. This produces two datasets that contain millions of pixels, enough to compute
reliable metrics. Second, during the extrapolation experiments, we divided the study area as
showed in figure 7; training and test datasets amount to millions of pixels in this case as well.
We reported performance metrics for both training area and test area to show the readers that
our models are not overfitting on the training set, which is a frequent problem of machine
learning algorithms.

Indeed, use of the training metrics instead of the test ones in the abstract results is a mistake

and we will correct it.

**Actions:** Useful reference for the ratio between the training and test sets has been inserted in Section 4 (lines 263-265), and the mistake in the abstract has been corrected (line 10).

10. **Reviewer 2 (ZH):** *Tables 1, 2, and 3 are not very clear. I would suggest bolding the important values in these tables.*
**Authors:** This suggestion is appreciated, we will make more readable the tables of the manuscript.

   **Actions:** As it is explained in point number 6, we improved metrics representation in the Tables.

11. **Reviewer:** *The investigated area was divided into four parts: A, B, C, and D. It seems that B and C divided the area into left and right parts. However, the choice of A and D is not very clear, and most areas of A and D overlap.*

   **Authors:** Exactly as stated in the point raised by the Reviewer, B and C divide the area into left and right parts, to examine model performance if the training considers just the upstream (area B for training, C for testing) or the downstream (area C for training, B for testing) portions of the Po river basin. Area A entirely encompasses the Alps and the streamline of the Po river, and part of the Po Plain; after training the models in A, we test them in the resting portion of the study area, which contains lower mountain range (the Apennines) and smaller river catchments. In this way, we check if our approach is sensitive to these hydrological conditions. Area D comprehends most of the Alps and the Po Plain, and entirely the Apennines and the streamline of the Po river. After using B for the training, we test the model over the resting part of the Alps and Po valley, checking if the approach is capable to estimate flood hazard in rivers with similar hydrological conditions to the ones of the training. We will improve the descriptions of these regions and why we selected them in our study.

   **Actions:** The description of the training and testing areas for extrapolation experiments have been improved in Section 4 (lines 269-277).

12. **Reviewer 2 (ZH):** *The authors used Gini importance (GI) to measure the importance of each factor. Is that possible to use the information to give different weights to the GDs to build up learning models?*

   **Authors:** Many thanks for raising this point on Gini Importance, which is a fundamental part of the description of our study outcome. We believe that GI is an extremely useful metric for guiding future studies on multivariate DEM-based flood hazard mapping. One might use in principle the GI values we obtained in our study to set initial values of weights for the input descriptors for training multivariate models in other study regions. Nevertheless, this represents a critical aspect as GI values resulted to be overly sensitive to the training area being used. We will include this comment in the revised manuscript.

**Actions:** A consideration about this issue is now present in the Discussion Section (lines 369-370).

13. ***Reviewer 3 (S-HY)****: The presentation of this manuscript is very clear. The authors provide six geomorphic descriptors to build decision tree model for classification and regression. By nature the DT is designed for calssification rather than regression. The authors need to give more details to describe how the DT can be used for regression.*

    **Authors:** we are very pleased to receive this overall positive evaluation of our manuscript, as well as to answer to this very important point. DTs are commonly used to address both classification and regression problems (see Hastie et al., 2009, https://doi.org/10.1007/978-0-387-84858-7), but indeed, their output always consists of a classification of the input data. This means that even if regression is performed, the continuous range of output values is binned into classes, and so the output of a regressor DT is a discrete estimation of the output variable. The number and distribution of the output classes for the predicted water depth depends on the tree structure (i.e., on the number of nodes and on the configuration), and it is automatically handled by the considered algorithm (i.e., function from software *scikit-learn*, see the manual on the official website https://scikit-learn.org/0.21/_downloads/scikit-learn-docs.pdf) once the tree parameters are decided by the user (maximum depth, minimum samples per leaf).
    In our case, the discretization of the output variable (i.e., water depth) is not a problem. In fact, as we use continuous flood hazard maps as target data and we have millions of observations (i.e., pixels), we can train very complex DTs, with many output classes for water depth. To give an idea of the grade of approximation, the regressor DT that produces the output map of figure 11 has about $10^4$ output classes.
    We will address these comments and make our description clearer in our revised manuscript.

    **Actions:** The use of decision trees for regression problems is now stated in Section 2 of the manuscript (line 137), while the good approximation of DTs for the regression problem is briefly commented in the Discussion (lines 387-388). We decided not to explain this with more detail in order to keep our work as straightforward as possible, but we provide some useful bibliography for interested readers (Breiman, 1984; Hastie et al., 2009).

14. ***Reviewer 3 (S-HY):*** *Figure 4 illustrates the water depth has been classifed into different groups <0.5, 0.5-1.0, 1.0-1.5 ...,>7. The reviewer guesses the regression was based on these grouping categories. More information is required*

    **Authors:** Thank you for pointing out this issue, which can lead to a misunderstanding of our results. As we stated in our reply to the Reviewer's previous comment, the classification of our regressor DTs is based on thousands of classes. The legend in Figure 4 (which is the same of figures 11 and 14) explains a classification of the raster values that has been chosen for an effective representation of the maps. We will clarify this aspect in the revised manuscript.

    **Actions:** A The legend of Figure 4 has been improved to specify that the showed colors solely refer to map visualization.

Moreover, we thankfully acknowledged your minor revisions, and changed our manuscript accordingly. Modifications on minor aspects are resembled in the following points:

- Typing errors have been corrected
- Figure 1 has been improved

Finally, we want to state again that we are very grateful for your appreciated help. We hope that the second version of our manuscript successfully addresses all your points.

Kind regards

Andrea Magnini, Michele Lombardi, Simone Persiano, Antonio Tirri, Francesco Lo Conti, Attilio Castellarin